# Dual-Sensor Signals Based Exact Gaussian Process-Assisted Hybrid Feature Extraction and Weighted Feature Fusion for Respiratory Rate and Uncertainty Estimations

**DOI:** 10.3390/s22218386

**Published:** 2022-11-01

**Authors:** Soojeong Lee, Hyeonjoon Moon, Mugahed A. Al-antari, Gangseong Lee

**Affiliations:** 1Department of Computer Engineering, Sejong University, 209 Neungdong-ro, Gwangjin-gu, Seoul 05006, Korea; 2Department of Artificial intelligence, Sejong University, 209 Neungdong-ro, Gwangjin-gu, Seoul 05006, Korea; 3Ingenium College, Kwangwoon University, 20 Kwangwoon-ro, Nowon-gu, Seoul 01897, Korea

**Keywords:** respiration rate estimation, exact Gaussian processing regression, hybrid feature extraction, weighted feature fusion, photoplethysmography, electrocardiogram

## Abstract

Accurately estimating respiratory rate (RR) has become essential for patients and the elderly. Hence, we propose a novel method that uses exact Gaussian process regression (EGPR)-assisted hybrid feature extraction and feature fusion based on photoplethysmography and electrocardiogram signals to improve the reliability of accurate RR and uncertainty estimations. First, we obtain the power spectral features and use the multi-phase feature model to compensate for insufficient input data. Then, we combine four different feature sets and choose features with high weights using a robust neighbor component analysis. The proposed EGPR algorithm provides a confidence interval representing the uncertainty. Therefore, the proposed EGPR algorithm, including hybrid feature extraction and weighted feature fusion, is an excellent model with improved reliability for accurate RR estimation. Furthermore, the proposed EGPR methodology is likely the only one currently available that provides highly stable variation and confidence intervals. The proposed EGPR-MF, 0.993 breath per minute (bpm), and EGPR-feature fusion, 1.064 (bpm), show the lowest mean absolute error compared to the other models.

## 1. Introduction

An increased respiratory rate (RR) over long periods indicates abnormal cardiorespiratory functions. Therefore, it is essential to estimate RR for cardiopulmonary health. The normal RR for adults at rest is 12–20 beats per minute (bpm) [1]. Currently, the capnograph is the standard instrument for measuring RR. However, such measurements are expensive and require strict management. The conventional respiration measurement methods used in clinical practice are unreliable [2,3]. Although most RR fluctuations are caused by body artifacts [4], non-contact respiration sensors are generally used to detect respiration. However, current contactless breathing measurements cannot account for this variability, as they widely use a single measurement value [5]. Alternatively, obtaining accurate RRs using a pulse oximeter, in addition to the oxygen saturation (SpO_2_) and heart rate (HR), is deemed patient-friendly and economical [6]. Today, researchers in biomedical and engineering fields are using wearable devices, which use photoplethysmography (PPG) signals to predict SpO_2_. Further, continuous signal measurement methods using PPG and electrocardiogram (ECG) sensors have been developed to estimate RR [7,8,9,10]. The wearable sensor-based ECG monitor can observe the patient’s HR and rhythm while walking [11]. PPG is also utilized for continuous HR observations in fitness devices and critically ill patients. PPG devices have previously been proposed for pulse transit time measurement and blood perfusion assessment [12]. Conventional wearable sensors and smartwatches can record PPG signals from one or more green, red, and infrared waveforms and measure the relative changes in the blood.

Machine learning (ML) is also widely used to estimate response values in the biomedical field. Lee et al. [13] proposed an approach for estimating RR using an ensemble-based gradient boosting algorithm (GBA) based on PPG signals; this approach depends on multiphase features [13] to improve the RR prediction capability. These features were extracted using several methods such as the autoregressive (AR) method [14], multifractal wavelet leaders (MWL) [15], wavelet packet [16], and maximal overlap discrete wavelet transform (MODWT) [17,18] to compensate for insufficient data. Additionally, RR estimation, including vital signals using a generative boosting long short-term memory network (LSTM), was proposed by Liu et al. [19]. Kumar et al. [20] introduced an algorithm for RR estimation using PPG and ECG signals based on the LSTM technique. This method is widely used in ML with time-series data and has particularly attracted attention in healthcare, such as for blood pressure and HR estimations based on PPG signals. Support vector regressions (SVR) assume a kernel function to determine the mapping between explanatory variables and response variables [21]. Artificial neural networks (ANNs) use a cost function to optimize the neural networks that determine the mapping between explanatory and response variables. Training sets are used to train models, while validation sets are used for tuning hyperparameters. Therefore, these models can be classified as parametric algorithms [21]. As an alternative technique, Gaussian process regression (GPR) [22,23] is an ML technique used for the classification of non-parametric models. The GPR model does not consider a particular parameter format for input–output mapping; instead, a Gaussian process is used beforehand to treat the input–output mapping as an arbitrary function with a defined probability density.

Furthermore, the training dataset is used to predict the parameters of this Gaussian distribution. Because GPR is robust to noisy data and naturally regularizes, it resists overfitting and produces uncertainty estimates [24]. The GPR can directly acquire model uncertainty, such as providing a distribution of the predicted values rather than a single value as the predicted value. This uncertainty cannot be obtained directly from SVR, ANNs, LSTM. However, an exact GPR (EGPR) model is inherently computer resource intensive [25]. Hence, this EGPR application was limited to a small training set of hundreds of data points [24]. In this study, we only have a small data set, which is another reason we can apply the EGPR model for RR estimation. The main advantage of the EGPR is that, like other kernel methods, given hyperparameter values (e.g., weight reduction and spreading of the Gaussian kernel), it can be optimized precisely. It is excellent, especially on limited datasets, because of its well-tuned smoothing and is still computationally reasonable. In addition, the EGPR comes with a straightforward way to tune hyperparameters by maximizing marginal possibilities. As a result, the EGPR consistently gives excellent fits without cross-validation.

Here, we propose a novel methodology to estimate RR with high confidence using EGPR-assisted hybrid feature extraction and feature fusion based on PPG and ECG (dual) signals. However, our limited data can lead to overfitting when using ML techniques [26]. In general, ML models trained on limited data, especially ANN and LSTM models, often exhibit unstable behavior in performance due to their sensitivity to initial parameter values and training order [27]. Hence, like all ML models, EGPR requires an adequately sized data set to train well. Hence, we obtain the input data dimension using the power spectral (PS) features based on an autocorrelation function and then crop the signal to increase the input data. We then use the multi-phases (MF) feature extraction model based on the AR method [14], MWL [15], wavelet packet [16], and MODWT [17,18] to compensate for insufficient input data. Thus, we obtain hybrid feature extraction such as PPG-based PS features, PPG-based MF features, ECG-based PS features, and ECG-based MF features. Then, we fuse four different feature sets and choose features with high weights using a robust neighbor component analysis (RNCA) [28]. The proposed EGPR algorithm provides a confidence interval (CI) representing the uncertainty (physiological variability). The proposed EGPR algorithm, which includes hybrid feature extraction and weighted feature fusion, is an excellent model for accurate RR estimation because it is resistant to overfitting and provides well-corrected prediction CIs [29]. This study provides a new methodology that can address all the mentioned limitations. Notably, this study contributes toward RR estimations as follows:The proposed EGPR algorithm with hybrid feature extraction and weighted feature fusion is an excellent model for accurate RR and uncertainty estimations.As far as we know, this is the first study of the EGPR-based feature fusion model that expresses uncertainty in RR estimation by estimating confidence intervals (CIs).The proposed EGPR model shows to be powerful in practice for quickly estimating the RR in hospitals and healthcare centers, featuring better prediction performance and lower estimated mean absolute errors, and standard deviations, CIs.

The study of this paper is composed as follows. In Section 2, the collection of ECG and PPG signals as shown in Figure 1, preprocessing for feature extraction. The proposed PPG and ECG (dual) signals based EGPR-assisted hybrid feature extraction and feature fusion is shown in Section 3. Section 4 represents the experimental results and statistical analysis. Finally, discussion and conclusions are denoted in Section 5 and Section 6. The block diagram of the proposed method is shown in Figure 1.

## 2. Dataset and Feature Extration

### 2.1. Collection PPG Signals

We use the dataset Beth Israel Deaconess Medical Center (BIDMC) randomly drawn from the MIMIC-II dataset [30]. The BIDMC dataset includes ECG, PPG signals, and impedance pneumography (IP) respiratory signals obtained from critically ill patients. A data set of approximately 480 second (s) in length was unified to 400 s to match a data set, and the sampling frequency (Fs) was 125 Hz, consisting of 53 records measuring patients aged 19–90 years. Reference RR values were calculated using two sets of annotations of individual breaths of impedance pneumography signals [31].

### 2.2. Preprocessing for Feature Extraction Processes

PPG signals are commonly used for estimating several bio-signals, such as the RR, HR, and blood pressure. Resampled wave signals are extracted using signal preprocessing techniques. In Step 1, the PPG signals are collected, as mentioned in Section 2. In Step 2, we eliminate high-frequency signals using a Kaiser window with a cutoff frequency of 35 Hz and a signal bandwidth of 3 dB; this helps clean and remove any potential noise from the PPG signals as shown Figure 2. In Step 3, each PPG signal is automatically segmented to multiple pulses using the adaptive incremental merge segmentation technique [32]. In Step 4, the fiducial points are identified and extracted from the peak and trough of the PPG signal. Figure 2a shows an example of the peak points on the PPG signal. We can similarly detect the trough points.

In Step 5, we resample irregular PPG signals at 5 Hz using linear interpolation [32]. In Step 6, we remove the low frequency signal using the Kaiser window with a cutoff frequency of 0.0665 Hz and a bandwidth of 3 dB. Finally, in Step 7, the resampled waveform is acquired from the PPG signals. The resampled wave dataset was collected from 53 different records or subjects. We estimated the RR value using the mean seconds between continuous breathing of the Hamming window. The proposed methodology provided better performance using a window size of 32 s rather than 16 or 64 s, as previously used by [30,31], respectively. The preprocessing of the ECG signals was conducted in almost the same method as the PPG signals.

## 3. Exact Gaussian Process Regression (EGPR) Based Hybrid Features Extraction and Weighted Feature Fusion

### 3.1. Multi-Phases (MF) Model for Feature Extraction

We used the wavelet transform to extract features and AR coefficients [14] using the segmented PPG signal. Specifically, the wavelet packet entropy is obtained using the MODWT model [17]. The MWL is acquired from a wavelet reader using an orthogonal spline wavelet filter [15]. We also used the 4-AR model order for RR estimation. Here, AR parameter coefficients are obtained using Burg’s model [14]. The key feature of this technology is that it extracts features by combining different technologies. In this study, we developed a parallel combination of AR model [14], MWL [15], wavelet packet entropy [16], and MODWT [17,18]. When using the BIDMC dataset, the 230 feature dimensions consisted of 40 AR features, 160 Shannon entropies, 20 fractal estimates using MWL, and 10 wavelet variance estimates as shown in Figure 3a. Interested readers can refer to [13].

### 3.2. Power Spectral Features Extraction

RR estimation is closely related to converting to bpm by multiplying the respiratory rate by 4 for more than 15 s [20]. For this reason, we can use the breathing frequency to obtain an automated feature vector from the power spectral based on the autocorrelation function. As mentioned above, signal-based features can be extracted from resampled wave signals obtained using the preprocessing step. The power spectral is an effective candidate for automatically extracting such features based on the autocorrelation function, as shown in Figure 3. The autocorrelation function measures the signal similarity between a given time axis and its delayed version over successive time intervals; this is expressed using the mean, variance, and covariance. The mean value is obtained from the input data set as $x={\left\{x_{n}\right\}}_{n=1}^{N}$, which specifies the expected value E[x] at each discrete time *n*; here, the mean function is denoted as $\mu_{x}=E\left[x\right]$. The autocorrelation function is based on the difference between discrete time *n* and n+m. If m=0 (delay 0), the autocorrelation function represents the maximum value, which can be described as the total energy as follows
(1)$$\nu_{m}\left(x\right)=\frac{\sum_{n=1}^{N}\left(x_{n}-\mu_{x}\right)\left(x_{n+m}-\mu_{x}\right)}{\sum_{n=1}^{N}{\left(x_{n}-\mu_{x}\right)}^{2}}$$

The power spectral denotes a fast Fourier transform of the correlation coefficient and provides information regarding the correlation structure of the wave signal [33]. The corresponding time-domain of the power spectral represents the autocorrelation because the time-domain autocorrelation function has the same formula characteristics and is equal to the square of the amplitude spectral. Thus, we acquire the power spectral of all the components within the range 0.1–2.5 Hz. Moreover, as mentioned in the preprocessing step, the length of each record was 400 s, and a sampling frequency of 125 Hz was used. Therefore, 50,000 experimental wave samples were prepared for each participant. Then, we obtain features using the several power spectra (PS) in Figure 3b. Finally, we stack the power spectral features for each record, respectively, as shown in Figure 3b. In detail, we rebuild the wave signals using the ECG and PPG signals 400 s and remake them into (12 × 32 windows). Then, we can acquire 12 × 256 (=3072) data points from long-resampled wave signal (400 s) in the BIDMC dataset. Hence, we obtain 12 × 53 (=636) samples with 256 dimensions of the feature, where 53 denotes the number of subjects. We illustrate the hybrid feature extraction process well, as shown in Figure 3.

### 3.3. Features Fusion

In this paper, we propose a feature fusion (FF) model using hybrid feature extraction, such as PPG-based PS features, PPG-based MF features, ECG-based PS features, and ECG-based MF features, to improve RR estimation reliability as shown in Figure 3. We combine a set of four types of features and then build a unified set of fused features into one. Although the constructed feature fusion set satisfies the input data dimension and number of samples, selecting features with high weights among the input features is necessary to improve the ML algorithm’s performance. In addition, we present an arithmetic fusion (AF) method that calculates the MAE results of the hybrid feature set to compare and evaluate the proposed feature fusion (FF) method.

### 3.4. Features Selection

As mentioned, we acquired a set of fuse features, resulting in higher-order features. However, we cannot guarantee that all features will provide helpful information to estimate RR. Also, fuse feature vectors can be overlapped. These overlapping feature vectors unnecessarily increase the complexity of the ML algorithm. As a result, overlapping feature vectors may not give satisfactory performance. Therefore, reducing the feature space size is essential while retaining only relevant features. In this study, we choose weighted feature vectors among high-dimensional features using robust neighbor component analysis (RNCA) [28]. A diagonal adaptation of neighbor component analysis (NCA) trains weighted feature vectors by minimizing a cost function that measures the mean leave-one-out regression cost over a training dataset. A training dataset is defined as $\mathrm{TD}=\{\left(x_{i},y_{i}\right),i=1,...,n\}$. Herein, we determined the weighted feature vector using the response vector y given the explanatory vectors x, where $x\in R^{p\times n}$ and *n* denote the number of observations. Regression was used to randomly select a reference point γ(x) in TD. Herein, we set the response variable at x to the response variable of the reference point γ(x) [28].
(2)$$D_{w}=\sum_{m=k}^{p}w_{k}^{2}\left|x_{ik}-x_{jk}\right|$$
where $D_{w}$ denotes the weighted distance function and $w_{k}$ is a weighted feature with *k*th feature. Hence, the probability $P(\gamma \left(x\right)=x_{j}|\mathrm{TD})$ that point $x_{j}$ is selected from T as the reference point:(3)$$P\left(\gamma \left(x\right)=x_{j}|\mathrm{TD}\right)=\frac{k(D_{w}\left(x_{i}-x_{j}\right))}{\sum_{j=1}^{n}k\left(D_{w}\left(x_{i}-x_{j}\right)\right)}$$
where (k(z)=exp(−z/σ)) is a kernel, and the kernel width σ denotes a parameter that influences the probability of each point being chosen as the reference point [28]. Here, we suppose that $P\left(\gamma \left(x\right)=x_{j}|\mathrm{TD}\right)\propto k\left(D_{w}\left(x_{i},x_{j}\right)\right)$. We predict the response to $x_{i}$ based on the trainning data set in $T^{-i}$, $\left(x_{i},y_{i}\right)$. The probability that $x_{j}$ is chosen as the reference point for $x_{i}$ is defined as
(4)$$\gamma_{ij=}P\left(\gamma \left(x\right)=x_{j}|{\mathrm{TD}}^{-i}\right)=\frac{k(D_{w}\left(x_{i}-x_{j}\right))}{\sum_{j=1,j\neq i}^{n}k\left(D_{w}\left(x_{i}-x_{j}\right)\right)}$$
(5)$$L_{i}=E\left(L\left(y_{i},{\hat{y}}_{i}\right)|{\mathrm{TD}}^{-i}\right)=\sum_{j=1,j\neq i}^{n}\gamma_{ij}L\left(y_{i},y_{j}\right)$$
where L is the cost function that measures the disagreement between $({\hat{y}}_{i}$, $y_{i})$. Hence, we include a regularization parameter λ to minimize the cost function as,
(6)$$F_{w}=\frac{1}{n}\sum_{i=1}^{n}L_{i}+\lambda \sum_{m=1}^{p}w_{m}^{2}$$

Here, we can use the regularization parameter λ(=0.001) to select the weighted feature vectors in high-dimensional features using the NCA algorithm [28] as shown (Equation 2) to (Equation 6).

### 3.5. Robust Neighborhood Component Analysis (RNCA)

The performance of the RNCA is highly dependent on the regularization parameter λ. The idea is to set the parameter λ to decide the best value for use in the RNCA model. Therefore, we can tune the regularization parameters using 5-fold cross-validation and mean squared error, as shown in Algorithm 1 and Figure 4. Herein, we used a custom robust cost function defined as $\zeta =1-\exp(-|y_{i}-y_{j}|)$. This function may be robust to outliers for use in the RNCA for the EGPR model. Thus, we tuned the parameter λ using the defined robust cost function. Next, we decided the λ value that produced the minimum average cost. Finally, we obtained the high-weighted feature vector without selecting other low-weighted features, as shown in Figure 4.

**Algorithm 1:** RNCA
**Procedure**: Input (${\mathrm{TD}}_{i=1}^{n}$): separate data set into training and testing sets
partition training data set into 5 folds**for**  
i=1,n  
**do**    $\lambda_{i,k}$: tuning using 5-fold cross-validation    **for** k=1,10 **do**        call NCA(TD,$\lambda_{i,k}$): train NCA for λ        compute $L_{i,k}$: record loss values    **end for**
**end for**
$L_{\mu }$ = mean($L_{i,k}$): compute average lossvalue$\lambda_{b}=\arg\min_{L_{\mu }}\left(y|x,\lambda_{i,k},L_{\mu }\right)$: find best $\lambda_{b}$call NCA(TD,$\lambda_{b}$, ζ): ζ = $@\left(y_{i},y_{j}\right)1-\exp(-|y_{i}-y_{j}\left|\right)$return (w) that produces weighted feature vectorsselect (w)≥ threshold
**End procedure**


### 3.6. EGPR Model for RR Estimation

EGPR is a flexible and robust non-parametric Bayesian algorithm for supervised ML [22]. The training input and output datasets are $D={\left\{x_{n},y_{n}\right\}}_{n=1}^{N}$ and $x\in R^{N\times D}$ and $y\in R^{N\times 1}$, respectively. To estimate y with the given x, the mapping function f=f(x) is used. According to these criteria, EGPR can be used as a non-parametric prior distribution of the mapping function [23]. Thus, we assume that the targets y can be obtained from the corresponding $x^{T}w$ by adding Gaussian noise as
(7)$$y=x^{T}w+\varepsilon ,\;\varepsilon ∽N\left(0,\sigma^{2}I\right)$$

The preprocessed signal wave dataset obtained the coefficient vectors (weights) w and variance $\sigma^{2}$. The EGPR model can estimate the response variables in Gaussian processes (GP) by including the mapping function f(x) and explicit basis functions θ as
(8)$$f\left(x\right)∽GP(0,c\left(x,x^{{}^{\prime }}\right))$$
where f(x) is obtained from a zero-mean EGP model with the covariance function $c(x,x^{{}^{\prime }})$ [23]. In practice, we obtain the mapping function as $f\left(x\right)=\theta {\left(x\right)}^{T}w$. The mean function of the input data denotes the expected value of the mapping function, μ(x)=E[f(x)]. The covariance function of the latent variable captures the smoothness of the response variable, and the basic functions project the input data x into the *p*-dimensional feature space:(9)$$c\left(x,x^{{}^{\prime }}\right)=E\left[\left(f\left(x\right)-\mu \left(x\right)\right){\left(f\left(x^{{}^{\prime }}\right)-\mu \left(x^{{}^{\prime }}\right)\right)}^{T}\right]$$

The expected value of (Equation 9) for the covariance function can be expressed as
(10)$$c\left(x,x^{{}^{\prime }}|\eta \right)\approx \sigma^{2}\exp\left(-\frac{{∥x-x^{{}^{\prime }}∥}^{2}}{2\eta^{2}}\right)$$
where c denotes a kernel function for EGPR [22] and η denotes a hyper-parameter. Here, we used exponential squares as the kernel function, as shown in (Equation 10). Hence, the kernel function can determine the properties of the mapping function f(x). Based on the EGPR algorithm, an instance of the target y can be defined as follows:(11)$$p\left(y_{n}|f\left(x_{n}\right),x_{n}\right)∽N\left(y_{n}|\theta {\left(x_{n}\right)}^{T}w+f\left(x_{n}\right),\sigma^{2}\right)$$
where $\theta (x_{n})$ is an instance of basis function which transform the original feature vectors x into a new feature vectors θ(x). Hence, we need $\Omega =\{w,\eta ,\sigma^{2}\}$ from the data set D and the marginal likelihood is defined as
(12)$$p\left(y|x\right)=p\left(y|x,\Omega \right)\approx N(y|\Theta w,c\left(x,x^{{}^{\prime }}|\eta \right)+\sigma^{2}I),$$

It is commonly used to train EGPR by finding local maxima for hyper-parameters Ω. The selection of the suitable kernel relies on the hypothesis such as the smoothness of the data and the expected pattern. In practice, we estimate the hyper-parameters Ω by maximizing the log marginal likelihood as
(13)$$\log p\left(y|x,\Omega \right)=-\frac{1}{2}\log\left|c\left(x,x^{{}^{\prime }}|\eta \right)+\sigma^{2}I\right|-\frac{1}{2}n\log2\pi \;\;\;\;\;\;\;\;\;\;\;\;\;\;\;\;\;\;\;\;\;\;\;\;\;\;-\frac{1}{2}{\left(y-\Theta w\right)}^{T}{\left[c\left(x,x^{{}^{\prime }}|\eta \right)+\sigma^{2}I\right]}^{-1}\left(y-\Theta w\right)$$
where $c(x,x^{{}^{\prime }}|\eta )$ denotes the covariance function matrix and Θ is the matrix of the explicit basis functions. The log marginal likelihood is expressed as a penalty fit scale and is maximized by a gradient-ascent using optimization technique. The hyper-parameters $\Omega =\{w,\eta ,\sigma^{2}\}$ based on EGPR algorithm is by maximizing the likelihood p(y|x) as a function for Ω.
(14)$$L\left(\hat{\Omega }\right)=\arg\max_{\Omega }\log\left(y|x,\Omega \right)$$

First, we computes $\hat{w}\left(\eta ,\sigma^{2}\right)$ to estimate the hyper-parameters, that maximizes the log likelihood function respect to w for given $\left(\eta ,\sigma^{2}\right)$ as
(15)$$\hat{w}\left(\eta ,\sigma^{2}\right)={\left\{\Theta^{T}{\left[c\left(x,x^{{}^{\prime }}|\eta \right)+\sigma^{2}I\right]}^{-1}\Theta \right\}}^{-1}\Theta^{T}{\left[c\left(x,x^{{}^{\prime }}|\eta \right)+\sigma^{2}I\right]}^{-1}y$$

Using known hyper-parameters, probabilistic estimations for the Bayesian EGPR model need the probability density function $p(y^{*}|y,x,x^{*})$. However, we can estimate the target y using the finite input new data $x^{*}$, and we use multivariate normal distributions with covariance matrices generated by the kernel to predict the output for these data. Hence, we define the conditional probability distribution as
(16)$$p\left(y^{*}|y,x,x^{*}\right)=\frac{p(y^{*},y|x,x^{*})}{p(y|x,x^{*})}$$

To obtain the joint density probability in the numerator, as shown in (16), it is necessary to use the mapping functions $f^{*}$ and *f* as follows:(17)$$\begin{array}{cc} p\left(y^{*},y|x,x^{*}\right)=\int \int p(y^{*},y,f^{*},f|x, & x^{*})dfdf^{*}\\ & =\int \int p\left(y^{*},y|f^{*},f,x,x^{*}\right)p\left(f^{*},f|x,x^{*}\right)dfdf^{*} \end{array}$$

Here, the EGPR model assumes that each target $y_{n}$ depends only on the corresponding latent variable $f(x_{n})$ and the feature vector $x_{n}$. A detailed derivation of (17) is presented in the Abbreviations.

Given y,x and hyperparameters Ω, the expected value of the estimation is
(18)$$\begin{array}{cc} E(y^{*}|y,x,x^{*},\Omega ) & =\theta {\left(x^{*}\right)}^{T}w+c\left(x,x^{{}^{\prime }}|\eta \right)\varphi \\ \; & =\theta {\left(x^{*}\right)}^{T}w+\sum_{n=1}^{N}\varphi_{n}c\left(x^{*},x_{n}|\eta \right) \end{array}$$
where $\varphi ={\left[c\left(x,x\right)+\sigma^{2}I\right]}^{-1}\left(y-\Theta w\right)$. In practice, we need an optimal point-estimation ${\hat{y}}^{*}$ using a loss function as follows:(19)$$E_{L}\left({\hat{y}}^{*}|x^{*}\right)=\int L\left(y^{*},{\hat{y}}^{*}\right)p\left(y^{*}|x^{*},D\right)dy^{*}$$

The goal is to obtain an estimate $y^{*}\approx {\hat{y}}^{*}$ and minimize the expected value of the specified loss function that $L(y^{*},{\hat{y}}^{*})$ is computed by minimizing between $y^{*}$ and ${\hat{y}}^{*}$ as
(20)$${\hat{y}}_{\mathrm{optimal}}|x^{*}=\arg\min_{{\hat{y}}^{*}}E_{L}\left({\hat{y}}^{*}|x^{*}\right)$$

Commonly used metrics to evaluate estimation accuracy are the mean absolute error (MAE). The MAE is the mean value of the sum of the absolute differences between the actual and estimated values. This study uses the MAE as a loss function, as given by L.

## 4. Experimental Results

The first thing to do was to extract resampled wave signals from dual signals. This process filtered out very high and low frequencies. Fifty-three long-resampled wave signals were used to design the proposed EGPR algorithm with hybrid feature extraction and weighted feature fusion process. These signals were randomly split into 80% for the training dataset and 20% for the testing dataset. Although mentioned in the previous section, we used the PPG signals 400 s to reconstruct the wave signal (12 × 32 windows) and get it back. We can then obtain 12 × 256 (=3072) data points from the long resampled wave signal (400 s) of the BIDMC data set. Finally, we acquired 12 × 53 (=636) samples with 256 dimensions of the feature, where 53 represents the number of patients. We calculated the reference RR from oral and nasal pressure signals based on the custom breath detection algorithm [31]. The parameter adjustments of the conventional and proposed methodologies using the PPG signals for each model are shown in Table 1. The parameter tunings were excluded when using the ECG signals to save space in the paper since it was nearly the same. The feature extraction, training, and testing times are computed using MATLAB ^®^2022 (The MathWorks Inc., Natick, MA, USA) [34] based on the dataset as shown in Table 2. As a result, the proposed power spectral (PS) based on the autocorrelation function requires lower computational times than the multi-phase feature extraction (MF) model [13] using the BIDMC dataset as given in Table 2.

We acquired the mean absolute error (MAE) and standard deviation (SD) results using hybrid feature extraction such as PPG-based PS features, PPG-based MF features, ECG-based PS features, and ECG-based MF features. Each hybrid feature set was used to input machine learning algorithms such as SVR, LSTM, GBA, and the EGPR models. Therefore, we obtain 16 different MAE and SD results as shown in Table 3. The ECG-based SVR with multi-phase feature extraction (SVR-MF) and ECG-based SVR with power spectral based on autocorrelation (SVR-PS) models are represented in Table 3. The results are acquired using the ECG-based LSTM with multi-phases feature extraction (LSTM-MF) and ECG-based LSTM with power spectral based on autocorrelation (LSTM-PS) [19] are shown in Table 3. The ECG-based GBA with power spectral based on autocorrelation (GBA-PS) is shown in Table 3. The results acquired using the ECG-based EGPR with multi-phase feature extraction (EGPR-MF) and ECG-based EGPR with power spectral based on autocorrelation (EGPR-PS) are illustrated in Table 3. We evaluate each hybrid methodology using the MAE and SD based on the RR estimation results as listed in Table 3. The MAE and SD results showed the mean values of the 30 experiments. Based on statistical experiments, as shown in Table 3 using the ECG signals, the MAE result (1.873) of SVR-MF indicates slightly better performance than these (1.993) of SVR-PS. The ECG-based LSTM-PS (2.058) technique performs better than the ECG-based LSTM-MF (2.357) technique, as in Table 3. The ECG-based GBA-MF (1.363) technique performs better than ECG-based GBA-PS (1.9), as shown in Table 3. The ECG-based EGPR-MF is also compared with ECG-based EGPR-PS, as shown in Table 3. The MAE results show that the proposed ECG-based EGPR-MF (1.152) has outstanding results compared to other machine learning models, as shown in Table 3. When using the PPG signals, the MAE of SVR-MF is slightly decreased compared to that of SVR-PS. The LSTM-MF model shows almost the same performance as the LSTM-PS model, as in Table 3. However, the GBA-MF model performs better than the GBA-PS model, as shown in Table 3. The proposed EGPR-MF (0.993) is also compared with EGPR-PS (1.511), as represented in Table 3. The MAE indicates that the proposed EGPR-MF technique is superior to the EGPR-PS as in Table 3.

Next, we display the MAE results obtained using the dual signals based hybrid feature extraction and arithmetic fusion (AF) and hybrid feature extraction and feature fusion (FF) as in Table 4. The MAE results obtained using the SVR-AF model (1.957) show slightly better performance than the MAE of the LSTM-AF model (2.150) and worse than the MAE of the GBA-AF model (1.571). In addition, the EGPR-AF model gives the best MAE results (1.313) as given in Table 4. The MAE results acquire using the SVR-FF model (1.920) show slightly better performance than the MAE of the LSTM-FF model (2.160) and worse than the MAE of the GBA-FF model (1.123). Moreover, the proposed EGPR-FF model gives the best MAE results (1.064), as shown in Table 4. We also obtain the CIs using ECG-based EGPR-MF, ECG-based EGPR-PS, PPG-based EGPR-MF, PPG-based EGPR-PS, and EGPR-FF to display the uncertainty of RR values as given in Table 5.

Figure 5a shows the MAE and SD results obtained for ECG-based SVR-MF, ECG-based SVR-PS, ECG-based LSTM-MF, and ECG-based LSTM-PS concerning the reference RR values as given in Table 3. Here, the first box indicates that the results of the ECG-based SVR-MF have a lower MAE than the results of the remaining three boxes. It can be seen that the results in Table 3 and the results in Figure 5a are precisely the same. As shown in Figure 5b, the MAE and SD results are obtained for ECG-based GBA-MF, ECG-based GBA-PS, ECG-based EGPR-MF, and ECG-based EGPR-PS concerning the reference RR values. Here, the third box represents that the ECG-based EGPR-MF have a lower MAE than those of ECG-based GBA-MF, ECG-based GBA-PS, and ECG-based EGPR-PS models. Figure 5b also well shows the results in Table 3.

We display the MAE and SD results acquired for PPG-based SVR-MF, PPG-based SVR-PS, PPG-based LSTM-MF, and PPG-based LSTM-PS concerning the reference RR values as illustrated in Figure 6a. The first box indicates that the results of the PPG-based SVR-MF have a slightly lower MAE than the results of the remaining boxes. Next, in Figure 6b, the MAE and SD results are acquired for PPG-based GBA-MF, PPG-based GBA-PS, PPG-based EGPR-MF, and PPG-based EGPR-PS concerning the reference RR values. Here, the first and third boxes represent that the results of the PPG-based GBA-MF and PPG-based EGPR-MF have lower MAEs than the results of the PPG-based GBA-PS and PPG-based EGPR-PS models.

As illustrated in Figure 7, we obtain the MAE results from the dual signals-based AF (a) and FF (b) models. Here, the MAE and SD result acquired from SVR-AF, LSTM-AF, GBA-AF, and EGPR-AF about the reference RR values. The last box represents that the proposed EGPR-AF has a lower MAE than the SVR-AF, LSTM-AF, and GBA-AF models, as shown in Figure 7a. The MAE and SD results are obtained using the SVR-FF, LSTM-FF, GBA-FF, AND EGPR-FF concerning the reference RR values. The last box represents the MAE results of the proposed EGPR-AF, which shows low MAE results among AF and FF methods, as shown in Figure 7b. Figure 8a displays the actual response (reference value), RR estimations, CI lower estimation, and CI upper estimation using the ECG-based EGPR-MF model. Figure 8b shows the actual response (reference value), RR estimations, CI lower estimation, and CI upper estimation using the EGPR-FF model.

### Statistical Analysis Using the ANOVA

We used ANOVA [35] experiments to compare and effectively evaluate the performance of the ECG-based SVR-MF, ECG-based SVR-PS, ECG-based LSTM-MF, and ECG-based LSTM-PS models. ANOVA is a commonly used statistical approach in situations where two or more population means are compared. In other words, the analysis of variance (ANOVA) has the following hypotheses of interest:(21)$$H_{0}:\mu_{1}=\mu_{2}...=\mu_{j},\;H_{1}:\mu_{1}\neq \mu_{2}...\neq \mu_{j}$$

The null hypothesis of ANOVA is that there is no difference in the mean. The alternative hypothesis is that the means are not the same. Therefore, we used multiple comparisons to determine different and average group results. One-way ANOVA denotes an easy example of a linear model, given that $e_{ij}=\alpha_{j}+\epsilon_{ij}$. Here, we assume that $e_{ij}$ is the experimental result (MAEs) of the four different groups as shown in Figure 5, where i(=30) is the number of experiments, and j(=4) is the number of groups. The performances of the ECG-based SVR-MF, ECG-based SVR-PS, ECG-based LSTM-MF, and ECG-based LSTM-PS models are determined based on the results of ANOVA experiments using the ECG signals. The corresponding within-group variation (error) and between-group variation (group) are shown in Table 6. MS is the mean squared error of 1.28, which is the SS/df ratio. The F-statistic (23.70) is the ratio of the mean squared error (1.28/0.054). The *p*-value of 5.55 $\times \;10^{-12}$ represents the probability that the test statistic acquires a value exceeding the calculated test statistic, which is P(F > 23.57). The small *p*-value, 5.55$\;\times \;10^{-12}$< (α=0.05), indicates that the differences between the group means are significant. The performances of the ECG-based GBA-MF, ECG-based GBA-PS, ECG-based EGPR-MF, and ECG-based EGPR-PS are also presented using the results of the ANOVA experiment. The MAEs of ECG-based GBA-MF, ECG-based GBA-PS, ECG-based EGPR-MF, and ECG-based EGPR-PS are significantly different, as shown in Table 6. The *p*-value of 1.82$\;\times \;10^{-50}$ is extremely lower than the significant value (0.05) in the last column of Table 6. Thus, it can be argued that the ECG-based EGPR-MF results show better accuracy than those of other models.

Using the PPG signals, we also compare the MAEs of all methodologies based on ANOVA experiments as shown in Table 7. The *p*-value of 0.037 is less than the significance value (0.05) in the 6th column of Table 7, which denotes that the differences between the group (models) means are significant. The results of the ANOVA experiment report that the performances of the PPG-based GBA-MF, PPG-based GBA-PS, PPG-based EGPR-MF, and PPG-based EGPR-PS are significantly different, as shown in Table 7. The *p*-value of 6.25$\;\times \;10^{-68}$ is extremely lower than the significant value (0.05) in the last column of Table 7. In Table 8, we also compare the MAEs of fusion models (AF) (four models) based on ANOVA experiments. The *p*-value of 3.45$\;\times \;10^{-67}$ is less than the significance value (0.05) in the 6th column of Table 8, which denotes that the differences between the group (models) means are significant. The results of the ANOVA experiment report that the performances of the SVR-FF, LSTM-FF, GBA-FF, and EGPR-FF models are significantly different, as shown in Table 8. The *p*-value of 3.85$\;\times \;10^{-63}$ is extremely lower than the significant value (0.05) in the last column of Table 8.

## 5. Discussion

Notably, this is the first study that combines EGPR with hybrid feature extraction and weighted feature fusion methodology to estimate RR and uncertainty from ECG and PPG signals. As shown in Table 2, we confirmed that the PS feature model is superior to the MF feature model in terms of computational complexity, which indicates that the PS model is more efficient than the MF model and saves computer resources. The MF feature method is a complex model that extracts features by combining several techniques, such as wavelet transform and entropy calculation. In contrast, the PS model has a simple architecture that uses power spectra based on the autocorrelation function. This study shows that the SVR-PS and EGPR-PS methods are very effective in computational complexity, whereas the LSTM-PS model is inefficient. This result will likely be available for using the proposed EGPR-PS in intelligent devices.

Furthermore, we noticed that using the dual signals (ECG and PPG), the proposed EGPR-MF model showed the lowest MAE results compared to the other models, such as SVR-MF, LSTM-MF, and GBA-MF. The EGPR-PS model also exhibited low MAE, 1.5 and 1.6 (bpm), and a very stable standard deviation (SD). We confirmed that the EGPR-AF model exhibited lower MAE than other methods showing simple mean results using PPG-based PS features, PPG-based MF features, ECG-based PS features, and ECG-based MF feature extraction models. In addition, we found that the proposed feature fusion (FF) model showed lower MAE results than the arithmetic fusion (AF) method while increasing the reliability of RR estimation. These results indicate that the proposed models are optimized to enable an exact balance between data fit and smoothness.

In addition, it shows that the proposed models are excellent algorithms because they are resistant to overfitting and provide well-corrected prediction errors based on small samples. Finally, we confirm that the fusion feature extraction process performed good respiration rate estimation. We predict confidence intervals (CIs) using ECG-based EGPR-MF, ECG-based EGPR-PS, PPG-based EGPR-MF, PPG-based EGPR-PS, and EGPR-based function methods to express uncertainty in RR estimation for the first time in the world. Table 5 showed the narrowest CI for the PPG-EGPR-MF model compared to other models. We demonstrated that the proposed EGPR-FF performed very well in estimating RR while estimating CIs to represent uncertainty in Figure 8. Respiratory rate (RR) measurement devices typically provide single-point predictions without CI. It does not offer a means to distinguish the statistical variation in the estimate from the variation in the estimate due to inherent variation due to physiology [36]. Therefore, predicting the CI for RR measures helps improve reliability.

Although we experimented with dual signals obtained from the BIDMC datasets [31], this study is limited due to the small number of samples with a relatively small number of participants. Hence, we should cross-validate using other bio-dataset. Furthermore, the proposed fusion method has high complexity because it consists of several types of feature extraction and fusion steps. Therefore, it is necessary to devise a method to alleviate the complexity in future studies. However, we will not claim that all our experiments are consistent with those described above. Also, algorithms are not detailed enough to replicate accurately in some cases.

## 6. Conclusions

In conclusion, we proposed a novel technique that uses PPG and ECG signals based on EGPR-assisted hybrid feature extraction and feature fusion to improve the reliability of accurate RR estimations. The EGPR, a non-parametric approach, provided a significantly better RR estimation accuracy and uncertainty estimation than parametric models such as SVM and GBA. First, we obtained the input data dimension using the power spectral (PS) features based on an autocorrelation function and then segmented the signal to increase the input data. We then used the multi-phases (MF) feature extraction model based on the AR method, MWL, wavelet packet, and MODWT to compensate for insufficient input data. Therefore, we acquired hybrid feature extraction such as PPG-based PS features, PPG-based MF features, ECG-based PS features, and ECG-based MF features. Then, we fused four different feature sets and chose features with high weights using a robust neighbor component analysis (RNCA). In addition, the proposed EGPR algorithm provides a confidence interval (CI) representing the uncertainty (physiological variability). Therefore, the proposed EGPR algorithm, including hybrid feature extraction and weighted feature fusion, is an excellent model with improved reliability for accurate RR estimation. In the future, we plan to increase the number of ECG and PPG records to improve our RR estimates, which vary widely across age groups and genders. Finally, the proposed model can be used to design a framework for improving telemedicine monitoring and optimizing clinical decision support frames.

## Figures and Tables

**Figure 1 sensors-22-08386-f001:**
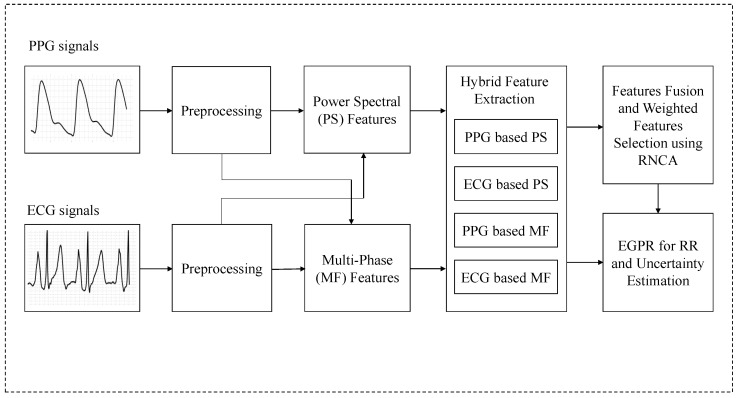
Block diagram of the proposed methodology using exact Gaussian process regression (EGPR) with hybrid feature extraction, feature fusion, and weighted features selection using the RNCA model.

**Figure 2 sensors-22-08386-f002:**
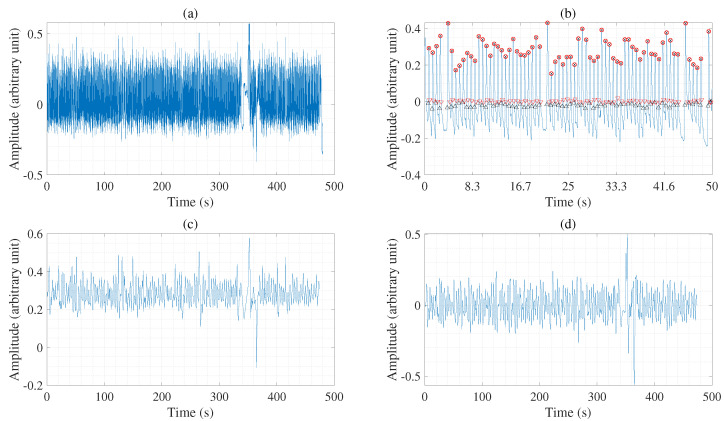
An example of preprocessing, where (**a**) denotes a PPG signal after removing a high-frequency (HF) signal; (**b**) denotes a partial PPG signal of 50 s shows maximum points in the ppg signal of about 480 s; (**c**) is maximum points in the ppg signal (480 s); (**d**) denotes a resample PPG signal.

**Figure 3 sensors-22-08386-f003:**
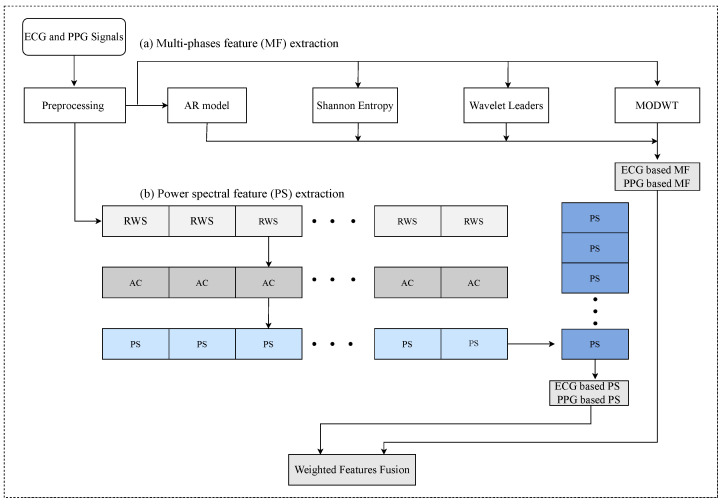
Hybrid feature extraction and weighted features fusion process, where (**a**) denotes the detailed process of parallel feature extraction using multi-phases (MF) model and (**b**) denotes power spectral (PS) feature extraction, where RWS denotes a segmented resample wave signal, AC is an autocorrelation for each record or subject.

**Figure 4 sensors-22-08386-f004:**
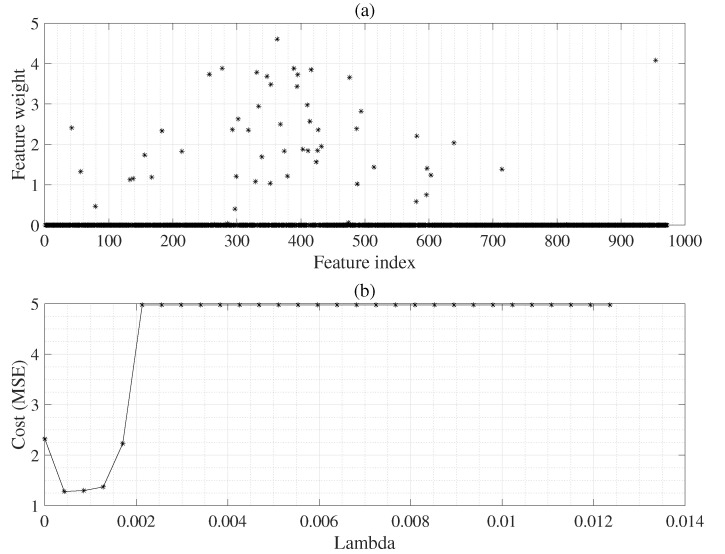
High weight features selection is performed using the RNCA model; where (**a**) is the weighted features and (**b**) is the regularization parameters λ.

**Figure 5 sensors-22-08386-f005:**
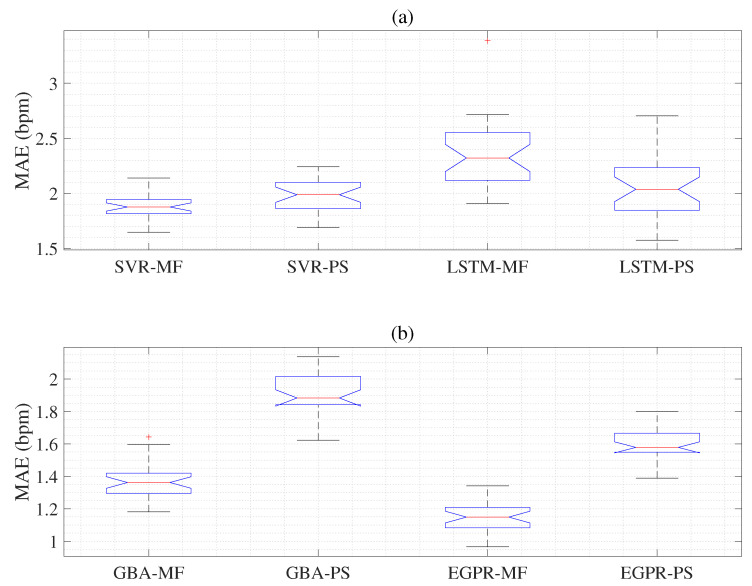
Top panel (**a**) represents the MAE and SD compared to the reference RR method acquired from the ECG-based SVR-MF, ECG-based SVR-PS, ECG-based LSTM-MF, and ECG-based LSTM-PS models. Bottom panel (**b**) shows the MAE and SD compared to the reference RR method obtained from the ECG-based GBA-MF, ECG-based GBA-PS, ECG-based EGPR-MF, and ECG-based EGPR-PS.

**Figure 6 sensors-22-08386-f006:**
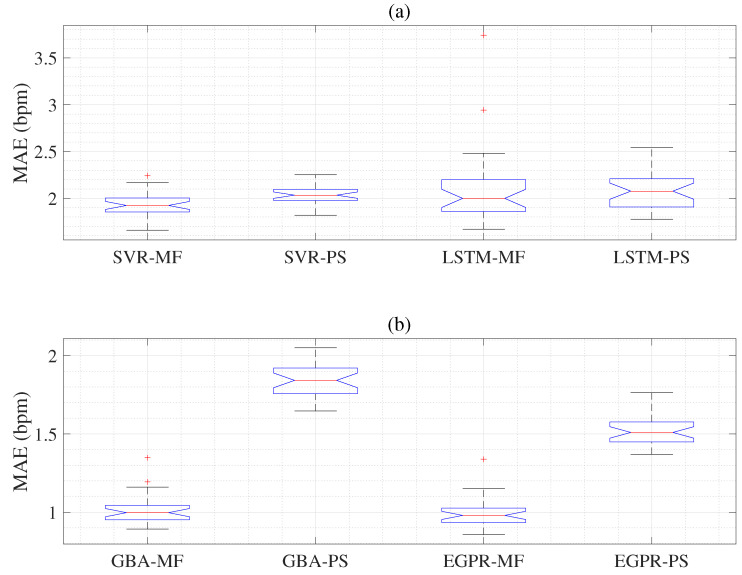
Top panel (**a**) denotes the MAE and SD compared to the reference RR method obtained from the PPG-based SVR-MF, PPG-based SVR-PS, PPG-based LSTM-MF, and PPG-based LSTM-PS models. Bottom panel (**b**) shows the MAE and SD compared to the reference RR method obtained from the PPG-based GBA-MF, PPG-based GBA-PS, PPG-based EGPR-MF, and PPG-based EGPR-PS.

**Figure 7 sensors-22-08386-f007:**
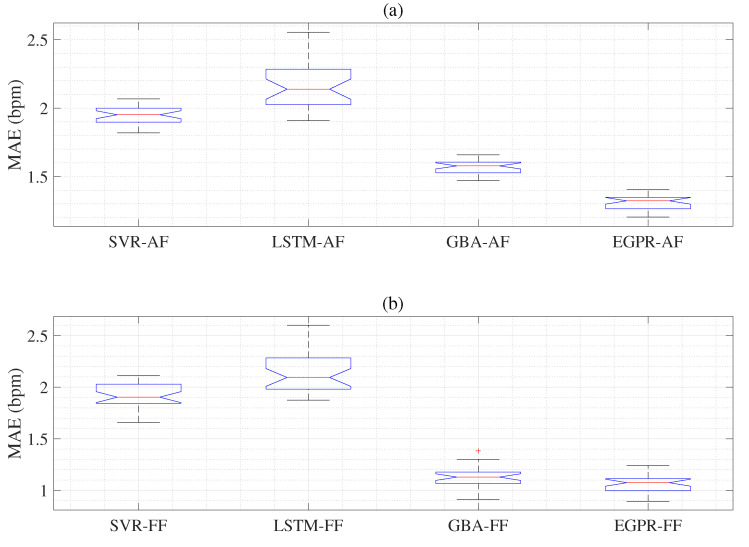
Top panel (**a**) denotes the MAE and SD compared to the reference RR method obtained using the dual signals based hybrid feature extraction and arithmetic fusion (AF) models., and hybrid feature extraction and feature fusion (FF) models. Bottom panel (**b**) shows the MAE and SD compared to the reference RR method obtained using the dual signals based hybrid feature extraction and feature fusion (FF) models.

**Figure 8 sensors-22-08386-f008:**
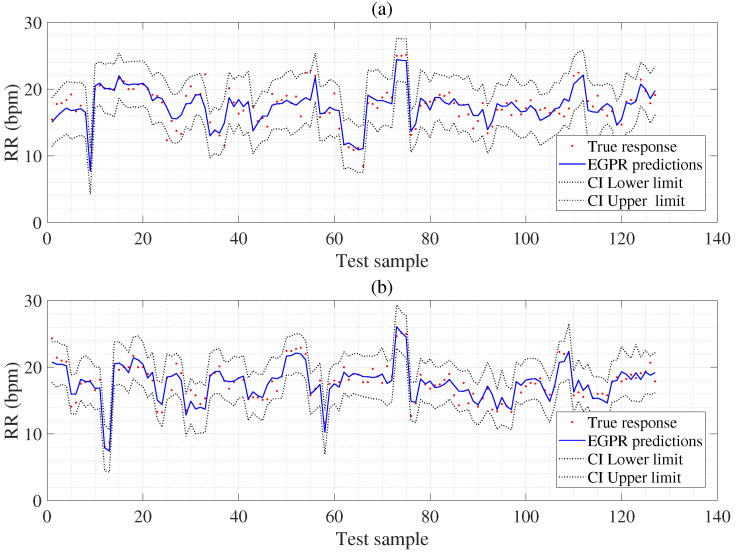
Top panel (**a**) denotes the CI estimation using the ECG-based EGPR-MF model to represent the uncertainty for RR estimation; bottom panel (**b**) is the CI estimation using the EGPR-FF model to represent the uncertainty for RR estimation.

**Table 1 sensors-22-08386-t001:** Summarized parameters of the proposed EGPR with hybrid features based on PPG signals, where Con. denoted a convolution kernel for the GBA method, SE was a squared exponential kernel for the EGPR algorithm.

Parameters	SVR-MF	SVR-PS	LSTM-MF	LSTM-PS	GBA-MF	GBA-PS	EGPR-MF	EGPR-PS
Input dimension	230	256	230	256	230	256	230	256
Output dimension	1	1	1	1	1	1	1	1
Hidden unit on the layers	-	-	200-300	200-300	-	-	-	-
Epsilon	3	3	1.00 $\times \;10{}^{-8}$	1.00 $\times \;10{}^{-8}$	-	-	-	-
Iterations	-	-	-	1000	1000	1000	-	-
FullyConnectdLayer	-	-	50	50	-	-	-	-
Dropout	-	-	50%	50%	-	-	-	-
MaxEpoch	-	-	200	200	-	-	-	-
GrandientThreshold	-	-	1	1	-	-	-	-
ShrinkageFactor	-	-	-	-	0.05–0.1	0.05–0.1	-	-
SubsamplingFactor	-	-	-	-	0.3	0.3	-	-
MaxTreeDepth	-	-	-	-	4	4	-	-
KernelFunction	Gauss.	Gauss.	-	-	Con.	Con.	SE	SE

**Table 2 sensors-22-08386-t002:** We compare feature extraction, training, and testing times between the traditional and proposed methodologies using Intel^®^Core(TM) i5-9400 CPU 4.1 GHz, OS 64 bit, RAM 16.0 GB, and MATLAB^®^2022 (The MathWorks Inc., Natick, MA, USA) system specifications.

Signals	SVR-MF	SVR-PS	LSTM-MF	LSTM-PS	GBA-MF	GBA-PS	EGPR-MF	EGPR-PS
ECG	21.74	1.75	86.86	73.75	23.30	3.31	22.10	2.11
PPG	21.04	1.99	84.64	71.38	22.60	3.55	21.40	2.35

**Table 3 sensors-22-08386-t003:** The RR estimation results obtained using the ECG and PPG-based all hybrid models are computed as the difference from the reference RR values to express it as the MAE and SD results.

Signals	ERROR	SVR-MF	SVR-PS	LSTM-MF	LSTM-PS	GBA-MF	GBA-PS	EGPR-MF	EGPR-PS
ECG	MAE	1.873	1.993	2.357	2.058	1.363	1.900	1.152	1.610
	SD	0.108	0.157	0.315	0.286	0.105	0.142	0.098	0.122
PPG	MAE	1.930	2.039	2.102	2.086	1.014	1.845	0.993	1.511
	SD	0.137	0.100	0.410	0.224	0.094	0.112	0.098	0.084

**Table 4 sensors-22-08386-t004:** The RR estimation results obtained using the dual signals based arithmetic fusion (AF) and feature fusion (FF) models are computed as the difference from the reference RR values to express it as the MAE and SD results.

Error	SVR-AF	LSTM-AF	GBA-AF	EGPR-AF	SVR-FF	LSTM-FF	GBA-FF	EGPR-FF
MAE	1.957	2.150	1.571	1.313	1.920	2.160	1.123	1.064
SD	0.066	0.152	0.051	0.051	0.115	0.226	0.103	0.082

**Table 5 sensors-22-08386-t005:** We compare the CIs obtained using the ECG-based EGPR-MF, ECG-based EGPR-PS, PPG-based EGPR-MF, PPG-based EGPR-PS, and EGPR based feature fusion (FF); where U denotes upper limit and L denotes lower limit.

RR (bpm)	ECG-EGPR-MF	ECG-EGPR-PS	PPG-EGPR-MF	PPG-EGPR-PS	EGPR-FF
RR (SD)	17.652 (2.451)	17.713 (2.435)	17.232 (2.776)	17.441 (2.538)	17.178 (3.167)
RR (SD) CI L	13.917 (2.539)	13.357 (2.592)	14.806 (2.772)	13.299 (2.680)	14.462 (3.123)
RR (SD) CI U	21.388 (2.435)	22.069 (2.399)	19.658 (2.789)	21.583 (2.635)	19.895 (3.240)
RR (SD) 95% CI	7.470 (0.844)	8.712 (1.109)	4.853 (0.330)	8.284 (1.577)	5.433 (0.600)

**Table 6 sensors-22-08386-t006:** ANOVA results of the left side columns are obtained from the ECG-based SVR-MF, ECG-based SVR-PS, ECG-based LSTM-MF, and ECG-based LSTM-PS models; ANOVA results of the right side columns are acquired from the ECG-based GBA-MF, ECG-based GBA-PS, ECG-based EGPR-MF, and ECG-based EGPR-PS, where SS is the sum of squares and df is the degrees of freedom. The total degree of freedom is 120 (=119-1) minus 1 from the total number of measurements (MAE).

Source	SS	df	MS	F	*p*-Value	SS	df	MS	F	*p*-Value
Group	3.83	3	1.28	23.70	5.55$\;\times \;10^{-12}$	9.24	3	3.08	250.1	1.82$\;\times \;10^{-50}$
Error	6.29	116	0.05			1.43	116	0.01		
Total	10.12	119				10.67	119			

**Table 7 sensors-22-08386-t007:** ANOVA results of the left side columns are obtained from the PPG-based SVR-MF, PPG-based SVR-PS, PPG-based LSTM-MF, and PPG-based LSTM-PS models; ANOVA results of the right side columns are acquired from the PPG-based GBA-MF, PPG-based GBA-PS, PPG-based EGPR-MF, and PPG-based EGPR-PS, where SS is the sum of squares and df is the degrees of freedom. The total degree of freedom is 120 (=119-1) minus 1 from the total number of measurements (MAE).

Source	SS	df	MS	F	*p*-Value	SS	df	MS	F	*p*-Value
Group	0.54	3	0.18	2.93	0.037	15.32	3	5.11	539.33	6.25$\;\times \;10^{-68}$
Error	7.16	116	0.06			1.10	116	0.01		
Total	7.70	119				16.42	119			

**Table 8 sensors-22-08386-t008:** ANOVA results of the left side columns are obtained from the dual signals based SVR-AF, LSTM-AF, GBA-AF, and EGPR-AF models; ANOVA results of the right side columns are acquired from the dual signals based SVR-FF, LSTM-FF, GBA-FF, and EGPR-FF models, where SS is the sum of squares and df is the degrees of freedom. The total degree of freedom is 120 (=119-1) minus 1 from the total number of measurements (MAE).

Source	SS	df	MS	F	*p*-Value	SS	df	MS	F	*p*-Value
Group	12.77	3	4.26	522.55	3.45$\;\times \;10^{-67}$	27.81	3	9.27	439.17	3.85$\;\times \;10^{-63}$
Error	0.95	116	0.01			2.45	116	0.02		
Total	13.72	119				30.26	119			

## Data Availability

Not applicable.

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
