# Peer review of "Dual-Sensor Signals Based Exact Gaussian Process-Assisted Hybrid Feature Extraction and Weighted Feature Fusion for Respiratory Rate and Uncertainty Estimations"

_sensors, 2022, doi:10.3390/s22218386_

Round 1
Reviewer 1 Report
In this work, the authors proposed a novel method based on Exact Gaussian Process Regression (EGPR) to estimate RR and uncertainty for photoplethysmography and electrocardiogram signals. The main motivation for using this type of algorithm instead of deep learning solutions is the limited amount of data in such scenarios.
The work is very interesting, and the solution is relevant, considering the importance of RR for cardiopulmonary health.
The authors can find below a list of suggestions to improve the overall quality of the manuscript.
- Concerning the abstract, it would be interesting if the authors included some information about the experimental results, so the readers could have a glance at the method's performance.
- The authors mentioned the limited amount of data to justify using GPR as an alternative. This is a very prevalent concern in medical applications in general. Consider including some references about this issue, such as Shaikhina, Torgyn, and Natalia A. Khovanova. "Handling limited datasets with neural networks in medical applications: A small-data approach." Artificial intelligence in medicine 75 (2017): 51-63.
- Overall the methods are properly detailed, with each step correctly exposed. However, there are some details that can be adjusted to improve this section. It would be interesting if the authors included a figure as an example of the effects of each preprocessing operation for a given example case. This would provide an easy visual assessment of the proposed preprocessing steps.
- In the experimental results section, the authors mentioned that the dataset split was 80% for training and 20% for testing. Since the dataset size is a major concern in this work, consider including the number of samples numerically to explicit this information.
- The authors also claim that computational complexity is one of the points to be observed in the proposed solution. In this way, please consider including information about the environment in which the method was performed, e.g., CPU and memory resources, etc.
- If possible, include some information about the execution time of the method.
- Regarding the manuscript text: Some references are missing, e.g., on page 2, lines 35 and 68.
- Also, some typos, such as the repeated "experimental results " in line 225 must be corrected. Please check the text to ensure there are no other errors.
- Finally, some paragraphs are excessively large. Please consider restructuring these large paragraphs with smaller ones to give readers a more pleasant experience.
Author Response
Response to the first reviewers’ comments
“Dual-Sensor Signals Based Exact Gaussian Process-Assisted Hybrid Feature Extraction and Weighted Feature Fusion for Respiratory Rate and Uncertainty Estimations”
Soojeong Lee, Hyeonjoon Moon, Mugahed A. AI, Gangseong Lee
Department of Computer Engineering, Sejong University, Seoul
Manuscript ID: sensors- 1978253
General
We appreciate the valuable comments and suggestions of the reviewers on our paper very much. In our resubmitted manuscript, we have incorporated all the comments and suggestions made by the reviewers and have given additional explanations. Our detailed responses are as follows.
- According to the comment: Concerning the abstract, it would be interesting if the authors included some information about the experimental results, so the readers could have a glance at the method's performance.
Answer 1, we included the experimental results as
“The proposed EGPR-MF, 0.993 breath per minute (bpm), and EGPR-feature fusion, 1.064 (bpm), show the lowest mean absolute error (MAE) results compared to the other models.”
page 1 (abstract)
- According to the comment: The authors mentioned the limited amount of data to justify using GPR as an alternative. This is a very prevalent concern in medical applications in general. Consider including some references about this issue, such as Shaikhina, Torgyn, and Natalia A. Khovanova. "Handling limited datasets with neural networks in medical applications: A small-data approach." Artificial intelligence in medicine 75 (2017): 51-63.
Answer 2, we included in our list of references Handling limited datasets with neural networks in medical applications: A small-data approach." as
“Shaikhina, T.; Khovanova, N.A. Handling limited datasets with neural networks in medical applications: A small-data approach. Artificial intelligence in medicine 2017, 75, 51-63.”
- According to the comment: Overall the methods are properly detailed, with each step correctly exposed. However, there are some details that can be adjusted to improve this section. It would be interesting if the authors included a figure as an example of the effects of each preprocessing operation for a given example case. This would provide an easy visual assessment of the proposed preprocessing steps.
Answer 3, we also included a figure as an example of the preprocessing step as Fig. 2
- According to the comment: In the experimental results section, the authors mentioned that the dataset split was 80% for training and 20% for testing. Since the dataset size is a major concern in this work, consider including the number of samples numerically to explicit this information.
Answer 4, we denoted the number of samples numerically to explicit this information as
“we used the PPG signals 400 s to reconstruct the wave signal (12 $\times$ 32 windows) and get it back. We can then obtain 12 $\times$ 256 (=3072) data points from the long resampled wave signal (400 s) of the BIDMC data set. Finally, we acquired 12 $\times$ 53 (= 636) samples with 256 dimensions of the feature, where 53 represents the number of patients.” Line 232-236.
- According to the comment: The authors also claim that computational complexity is one of the points to be observed in the proposed solution. In this way, please consider including information about the environment in which the method was performed, e.g., CPU and memory resources, etc.
If possible, include some information about the execution time of the method.
Answer 5, we represented information of the system. Also, we included information about the execution time “The feature extraction, training, and testing times are computed using MATLAB 2022 [34] based on the dataset as shown in Table 2.”
Caption in Table 2, “We compare feature extraction, training, and testing times between the traditional and proposed methodologies using Intel Core(TM) i5-9400 CPU 4.1 GHz, OS 64 bit, RAM 16.0 GB, and Matlab 2022 (The MathWorks Inc., Natick, Ma, USA) system specifications.”
- According to the comment: Regarding the manuscript text: Some references are missing, e.g., on page 2, lines 35 and 68.
Answer 6. We fixed it.
- According to the comment: Also, some typos, such as the repeated "experimental results " in line 225, must be corrected. Please check the text to ensure there are no other errors.
Answer 7, we removed some typos!
- According to the comment: Finally, some paragraphs are excessively large. Please consider restructuring these large paragraphs with smaller ones to give readers a more pleasant experience.
Answer 7, we separated the large paragraphs as lines 246-309

Reviewer 2 Report
Please see the attachment.

Author Response
Response to the second reviewers’ comments
“Dual-Sensor Signals Based Exact Gaussian Process-Assisted Hybrid Feature Extraction and Weighted Feature Fusion for Respiratory Rate and Uncertainty Estimations”
Soojeong Lee, Hyeonjoon Moon, Mugahed A. AI, Gangseong Lee
Department of Computer Engineering, Sejong University, Seoul
Manuscript ID: sensors- 1978253
General
We appreciate the valuable comments and suggestions of the reviewers on our paper very much. In our resubmitted manuscript, we have incorporated all the comments and suggestions made by the reviewers and have given additional explanations. Our detailed responses are as follows.
- According to the comment: 1) Abstract section:I believe the methods used can be more succinct, and add some background or purpose of research would be more attractive.
Answer 1, we revised the summary section based on comments from the second reviewer as
“ Accurately estimating respiratory rate (RR) has become essential for patients and the elderly. Hence, we propose a novel method that uses exact Gaussian process regression (EGPR)-assisted hybrid feature extraction and feature fusion based on photoplethysmography and electrocardiogram signals to improve the reliability of accurate RR and uncertainty estimations. First, we obtain the power spectral features and use the multi-phase feature model to compensate for insufficient input data. Then, we combine four different feature sets and choose features with high weights using a robust neighbor component analysis. The proposed EGPR algorithm provides a confidence interval representing the uncertainty. Therefore, the proposed EGPR algorithm, including hybrid feature extraction and weighted feature fusion, is an excellent model with improved reliability for accurate RR estimation. Furthermore, the proposed EGPR methodology is likely the only one currently available to provide highly stable variation and confidence intervals. The proposed EGPR-MF, 0.993 breath per minute (bpm), and EGPR-feature fusion, 1.064 (bpm), show the lowest mean absolute error compared to the other models. ”
page 1 (abstract)
- According to the comment: 2) Page 1, Line 3, “Accurate RR” appeared for the first time in the abstract, where it should be spelled out.
Answer 2, we fixed it as “Accurately estimating respiratory rate (RR) has become essential for patients and the elderly.”
- According to the comment: 3) Page 2 Line53-55, The description of the kernel function is inaccurate. ANN using neural network architecture to determine the mapping between explanatory variables and response variables, and use a “lose function” to optimize the neural networks.
Answer 3, we revised the sentence as “Support vector regression (SVR) assume a kernel function to determine the mapping between explanatory variables and response variables \cite{rakoto}. Artificial neural networks (ANNs) use a cost function to optimize the neural networks that determine the mapping between explanatory and response variables. Training sets are used to train models, while validation sets are used for tuning hyperparameters.” Lines 48-52.
- According to the comment: 4) Page 2 Line55-56, Training sets were used to train models (weights of the model), while validation sets were used for tuning hyperparameters.
Answer 4, we revised the sentence “Training sets are used to train models, while validation sets are used for tuning hyperparameters.” Lines 51-52.
- According to the comment: 5) Page 2 Line76-77, “Generally , limited input data cannot guarantee the successful performance of ML algorithms, including the ML and LSTM algorithms, due to many nonlinear functions”. This sentence is hard to read.
In answer 5, we revised the sentence as “However, our limited data can lead to overfitting when using ML techniques [26].” Line 76.
- According to the comment: 6) Page 3, Line 97-99: Authors should pay attention to the writing style of scientific papers.
Answer 6. We revised the sentence as “As far as we know, this is the first study of the EGPR-based feature fusion model that expresses uncertainty in RR estimation by estimating confidence intervals (CIs).” Lines 96-97.
- According to the comment: 7) Page 4, Line 118, ”BP” appeared for the first time, so it should be spelled out.
Answer 7, we fixed it as “PPG signals are commonly used for estimating several bio-signals, such as the RR, HR, and blood pressure.” Line 117.
- According to the comment: 8) Page 4, Line 137: Wavelet transform algorithm,AR model are lake of detail.
Answer 8, we included some information about the wavelet transform algorithm and AR model as
“We used the wavelet transform to extract features and AR coefficients [14] using the segmented PPG signal. Specifically, the wavelet packet entropy is obtained using the MODWT model [17]. The MWL is acquired from a wavelet reader using an orthogonal spline wavelet filter [15]. We also used the 4-AR model order for RR estimation. Here, AR parameter coefficients are obtained using Burg's model [14].” Lines 137-141.
- According to the comment: The improvements of EGPR model needs to be emphasized.
Answer 9, we added the sentence with respect to the EGPR model as “In this study, we only have a small data set, which is another reason we can apply the EGPR model for RR estimation. The main advantage of the EGPR is that, like other kernel methods, given hyperparameter values (e.g., weight reduction and spreading of the Gaussian kernel), it can be optimized precisely. It is excellent, especially on limited datasets, because of its well-tuned smoothing and is still computationally reasonable. In addition, the EGPR comes with a straightforward way to tune hyperparameters by maximizing marginal possibilities. As a result, the EGPR consistently gives excellent fits without cross-validation.” Lines 66-73.
- According to the comment: 10) Some references in the text do not have numbers.
Answer 10, we fixed it.
- According to the comment: The source code and some hyper-parameters is not provided - this method has the potential to be easily replicated, yet without the code, this would be a challenge.
Answer 11, Upon a reasonable request, the corresponding author can offer a partial code for the study upon completion of all projects. Line 425.
